# FEDERATED INFERENCE THROUGH ALIGNING LOCAL REPRESENTATIONS AND LEARNING A CONSENSUS GRAPH

## ABSTRACT

Machine learning is faced with many data challenges when applied in practice. Among them, a notable barrier is that data are distributed and sharing is unrealistic for volume and privacy reasons. Federated learning is a recent formalism to tackle this challenge, so that data owners can develop a common model jointly but use it separately. In this work, we consider a less addressed scenario where a datum consists of multiple parts, each of which belongs to a separate owner. In this scenario, joint efforts are required not only in learning but also in inference. We study *federated inference*, which allows each data owner to learn its own model that captures local data characteristics and copes with data heterogeneity. On the top is a federation of the local data representations, performing global inference that incorporates all distributed parts collectively. To enhance this local–global framework, we propose aligning the ambiguous data representations caused by arbitrary arrangement of neurons in local neural network models, as well as learning a consensus graph among data owners in the global model to improve performance. We demonstrate effectiveness of the proposed framework on four real-life data sets including power grid systems and traffic networks.

## 1 INTRODUCTION

Machine learning models become increasingly data hungry as the promise of deep learning continues to realize. More and more applications grow in scale thanks to the availability of distributed data across devices and organizations. Federated learning (McMahan et al., 2017; Yang et al., 2019; Li et al., 2019; Kairouz et al., 2019) emerges as a formalism that allows data owners to collaboratively train a common model by using one's own data without sharing. Such a formalism is poised to resolve the challenges of expensive data communication and the risk of privacy violation, in light of policies such as the General Data Protection Regulation (Albrecht, 2016).

An issue less addressed by federated learning is the inference process. In fact, inference therein is trivial: once the common model is learned, each data owner retains a copy and applies it on local data, independently of other owners. However, such a scenario is not the only one how data are distributed in practice. In this work, we consider the following scenario: a datum has multiple parts, each of which belongs to a separate owner. Then, the inference must be collectively performed by all participating owners, since none of them alone possesses the entire information.

Vertical federated learning studies such a scenario (Hardy et al., 2017; Hu et al., 2019; Chen et al., 2020). This concept is figuratively named through cutting the data matrix vertically along the feature axis, rather than the data axis. From the sporadic literature addressing in this scenario, methods generally introduce model parameters distributed with data parts and optionally global parameters that reside in a central server. All parameters are learned jointly, causing however a practical drawback—expensive coordination (even synchronization) is required among data owners and the central server (if present).

Consider a live example—the national electricity grid, over which thousands of phasor measurement units (PMUs) have been deployed to monitor the grid condition (Smartgrid.gov). PMU measurements, as time series data, are owned by several parties. These data may be used to train machine learning models that identify grid events (e.g., fault, oscillation, and generator trip). Such an event

detection system relies on collective series measurements at the same time window but distributed across different data owners. To minimize coordination among owners and maximize autonomy, *it is more desirable if each maintains a model of their own and does not participate joint training.*

In this work, we propose a local–global model framework that maintains data owner autonomy while staying effective in global inference. Therein, each data owner trains a local model with its data part; the training is independent and incurs no coordination. In the deep learning terminology, the local models produce data representations for input data. Then, a central server takes these representations as input and trains a model for global inference.

We term the scenario where inference is collectively performed by data owners but training incurs little coordination among them, *federated inference*, to distinguish federated learning and vertical federated learning. The local–global framework we propose for this scenario, however, bears two technical challenges. One is the ambiguity of local data representations, because feature dimensions can be arbitrarily permuted without changing the local model. Another challenge is how the global model leverages innate interactions of local data missed by independent local models.

We resolve the first challenge through aligning the feature dimensions across all local representations. We resolve the second challenge through employing graph neural networks as the global inference model, where the graph corresponds to the explicit or implicit relational structure of the data owners. When such a graph is not present, we treat the combinatorial graph structure as a random variable of the Bernoulli distribution and optimize the distribution parameters as well.

We summarize the contributions of this work as follows.

1. We formalized federated inference, a less addressed scenario of machine learning with distributed data, where inference is conducted jointly by data owners without data sharing and coordinated training (Section 2).

2. We propose an inference framework that consists of autonomous local models and a central model that digests local data representations and produces a global output (Section 3).

3. We address the ambiguity challenge of this framework through aligning local representations (Section 4) and address the missing of local model interactions through employing a graph neural network in the central global model. We further propose to simultaneously learn the graph structure if not present (Section 5).

4. We study approaches to latent alignment and approaches to graph structure learning and develop theoretical insights into these approaches (Theorems 1–3). As a byproduct, a more efficient Bernoulli sampling method **icdf** is proposed to sample graphs for structure learning.

5. We demonstrate experiments with four real-life data sets including power grids and traffic networks and show the effectiveness of the proposed framework (Section 6).

## 2 PROBLEM SETTING

Federated inference refers to a machine learning scenario where both training and inference are conducted on distributed data collectively by data owners. Each owner enjoys data and model autonomy but is subject to centralized coordination that produces a global prediction. This scenario stands in contrast to federated learning, whose inference process is local and separate among owners.

Formally, we use a superscript $i$ to index data owners. Let $x$ be a datum with label $y$ and let $x^i$ be the part of datum the $i$th owner possesses; that is, $x = (x^1, x^2, \ldots, x^n)$ with $n$ owners. The problem is to learn a model

$$y = f(x) = f(x^1, x^2, \ldots, x^n) \tag{1}$$

collectively with all data parts and to perform inference jointly by all owners. Additionally, owners share neither data nor models with each other for, e.g., privacy reasons. Moreover, owners do not participate joint training, which often incurs expensive coordination.

**Similarity to federated learning.** Federated inference shares the defining data characteristics of federated learning, first coined in McMahan et al. (2017): distributed, non-IID, and unbalanced. Data are distributed among owners but not shared. In fact, data may even be heterogeneous. For example, time series measures of the power grid may have different attribute dimensions and may be under different sampling frequencies. As a result, data size may vary significantly among owners.

**Dissimilarity to federated learning.** The root of the differences between these two concepts is the constituent of one datum (data point). In federated learning, a data point is the basic unit of data and thus all owners learn a common model but use it separately. On the other hand, federated inference is concerned with data split in parts across owners. All parts of a data point contribute to the inference collectively.

**Similarity and dissimilarity to vertical federated learning.** Both concepts are concerned with the split of a datum across owners. However, approaches taken for vertical federated learning differ substantially from ours because of joint training among owners. From the less prolific literature, two lines of work are noted. One takes the data matrix literally, by assuming tabular data and studying linear models, where model parameters have natural correspondence to the data parts (Hardy et al., 2017; Nock et al., 2018; Heinze et al., 2014; 2016). Often, these approaches are hard to generalize to complex data and/or many owners. Another line of work uses a local–global model framework similarly as we do but jointly trains these two parts, incurring expensive communication and creating dependence of local models (Hu et al., 2019; Chen et al., 2020).[1] In contrast, we allow data owners to train their models independently, maintaining local model autonomy.

**Data example.** Let us consider the power grid. Figure 1 pictorially illustrates PMU measurements distributed across data owners. A panel of time series corresponds to a specific time window and the series collectively represent one data point, which the event detection system classifies. In this simplified illustration, each data owner possesses one series recorded by one PMU; but in practice they may own different amounts of PMUs (and thus series). Moreover, the series may differ in length because of varying sampling frequencies; and the series are multivariate with possibly different number of variates. All these variations contribute to data heterogeneity, which necessitates the construction of separate local models. Note that if an event does not cascade over the entire grid, some local models may report event whereas others report normal, resulting in conflicting opinions. A consensus (global) model is responsible to resolve the conflict. Additionally, missing data may occur.

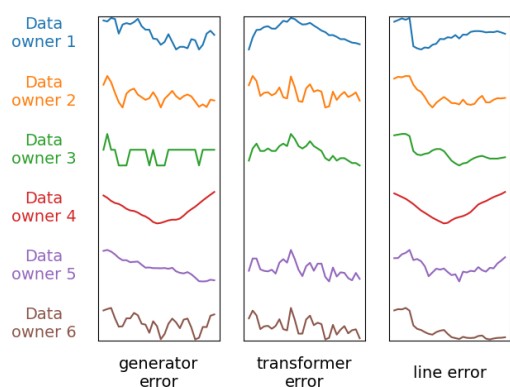

Figure 1: Federated inference: A global label is predicted collectively based on local data from multiple owners. Local data may be heterogeneous and missing data may occur.

# 3 FEDERATED INFERENCE FRAMEWORK

As such, the proposed framework for federated inference consists of local models $f^i$ and a global model $g$, such that their composition is the sought $f$ denoted in (1). Each data owner $i$ possesses a local model trained with its data, independently of other owners. This way, no data sharing is invoked and privacy is of minimal concern. However, because the local models lack a global vision and may be conflicting, a central (global) model is key to coordinating the local opinions for final prediction. To maintain autonomy, local models are frozen once pretrained and will not join the training of the global model. Data owners send local data representations to a centralized server for global model training (and inference). In other words, the global model queries neither the raw data nor the local models from data owners. As long as owners agree to send the less decipherable representations to the central server, global inference can be made.

**Local models.** We treat a neural network except the final output layer as a feature extractor, which produces the representation $h^i$ of an input $x^i$; and treat for simplicity the output layer as a logistic regression. That is, a local model $f^i$ reads:

$$f^i(x^i) = \text{softmax}(W^i h^i + b^i) \quad \text{where} \quad h^i = \text{embedding}(x^i). \tag{2}$$

---

[1]Note that the approach proposed by Hu et al. (2019) assumes no parameters for the global model. Were global parameters present, gradient communication is inevitable.

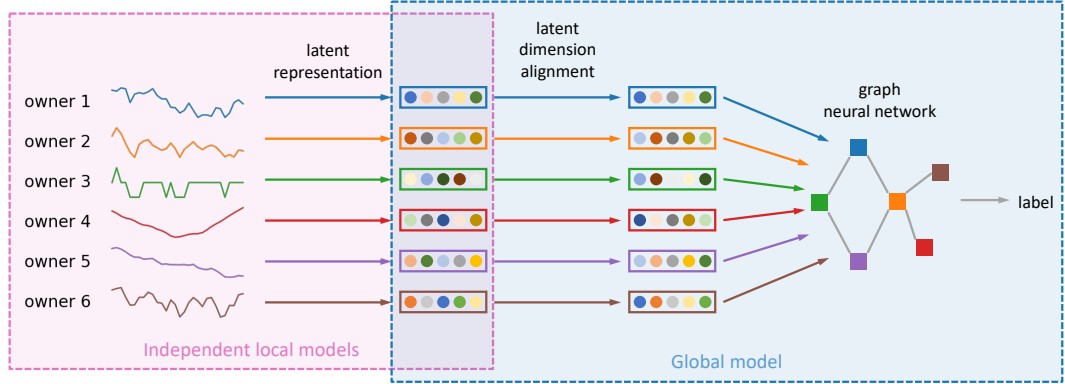

Figure 2: Federated inference framework. Local models are trained independently and separately from the global model. The algorithm is summarized in Algorithm 1 in supplement Section C.

We interchangeably use "representation", "embedding", and "latent vector" to mean $h^i$. These $h^i$'s are assumed to have the same shape across $i$, although $x^i$ can have different shapes and the embedding function can have different architectures to cope with data heterogeneity. A simple example of the embedding function is a fully connected layer $h^i = \text{ReLU}(U^i x^i + c^i)$; but an arbitrarily complex function is applicable.

**Global model.** The global model is a function $g$ of all local representations:

$$\widehat{y} = g(h^1, h^2, \dots, h^n). \tag{3}$$

An example of $g$ is a fully connected layer, followed by mean pooling and another fully connected layer:

$$\widehat{y} = \text{softmax}\left(W_1 \cdot \tfrac{1}{n} \sum_{i=1}^{n} \text{ReLU}(W_0 h^i + b_0) + b_1\right). \tag{4}$$

**Challenges.** Two considerations are pertinent to this framework. First, when the latent dimensions have semantic meaning (e.g., when the local models are trained to yield disentangled representations (Higgins et al., 2018)), each latent feature of the local representations may not match, because an arbitrary permutation of the latent dimensions does not change a local model. Second, a naive mean pooling as in (4) may miss the interdependencies between local data, leading to a less well performing global model. Such interdependencies naturally occur in the power grid example because of the physics of an electricity network. Hence, in subsequent sections, we use latent alignment to address the first problem and graph neural network to address the second one. Incorporating these two components, we show the full, proposed framework in Figure 2 and Algorithm 1 (supplement Section C).

## 4 ALIGNING LOCAL REPRESENTATIONS

For the global model to be meaningful, the feature dimensions of the local representations $h^i$ should match. For example, in (4), all $h^i$'s multiply the same weight matrix $W_0$; in other words, each element of $h^i$ corresponds to one input neuron of the initial fully connected layer. Permutations of the elements will destroy the correspondence. That is, even if the local models are fixed, the arbitrary arrangement of the feature dimensions of the latent vectors causes ambiguity of what an optimal global model can be built.

Mathematically, let us use a vector $\text{p}$ to denote permutation and place a superscript $i$ whenever necessary. The $i$th local model (2) can be equivalently written as

$$f^i(x^i) = \text{softmax}(W^i[:, \text{p}^i]h^i[\text{p}^i] + b^i) \quad \text{where} \quad h^i[\text{p}^i] = \text{embedding}(x^i; \text{p}^i), \tag{5}$$

for any permutation $\text{p}^i$, as long as the embedding function is able to produce a permuted $h^i[\text{p}^i]$ under the same input $x^i$. Such a requirement can be easily satisfied if the embedding function is a fully connected layer (i.e., $h[\text{p}] = \text{ReLU}(W[\text{p}, :]x + b[\text{p}])$). In fact, it is satisfied by most neural networks as well. In the supplement, we give another example: the GRU (Cho et al., 2014).

Hence, we propose to align the feature dimensions across all local vectors $h^i$ to disambiguate the ambiguity. This proposal amounts to modifying the global model (3) to the following:

$$\widehat{y} = g(P^1 h^1, P^2 h^2, \dots, P^n h^n), \tag{6}$$

where $P^i$ is an alignment matrix for each data owner $i$.

Two approaches of defining $P^i$ exist. The first approach is a soft alignment, which treats each $P^i$ a free parameter matrix to optimize. It may be square or rectangle, the latter case indicating a change of the number of features.

The second approach is a hard alignment, which treats each $P^i$ a permutation matrix. Learning permutation matrices is challenging, however, because they correspond to combinatorial structures and are unsuitable for gradient-based training. We follow Mena et al. (2018); Emami & Ranka (2018) and relax $P^i$ by a doubly stochastic matrix, which can be differentiably parameterized by the Sinkhorn–Knopp algorithm (Sinkhorn & Knopp, 1967). Specifically, starting from a nonnegative square matrix $K_0$ and column vectors $r_0 = c_0 = \mathbf{1}$ of matching lengths, define the sequence

$$c_{j+1} = \mathbf{1} \oslash (K_0^T r_j) \text{ and } r_{j+1} = \mathbf{1} \oslash (K_0 c_j), \quad \text{for } j = 0, 1, \dots \tag{7}$$

Then, under a mild condition, $K_j := \operatorname{diag}(r_j) K_0 \operatorname{diag}(c_j)$ converges to a doubly stochastic matrix. We truncate the sequence at the $T$th step and treat $K_T$ as an approximation of $P^i$.

Despite the advocation by Mena et al. (2018); Emami & Ranka (2018), we obtain the following convergence result of Sinkhorn–Knopp, which reveals no free lunch.

**Theorem 1** (informal). *Under a condition of $K_0$, there exists a positive integer $J$ and a constant $C_J$ such that for all $j \geq J$,*

$$\left\| \begin{bmatrix} K_j^T \mathbf{1} \\ K_j \mathbf{1} \end{bmatrix} - \begin{bmatrix} \mathbf{1} \\ \mathbf{1} \end{bmatrix} \right\| \leq C_J (1 + \sigma_2^2) \sigma_2^{2(j-J)},$$

*where $\sigma_2 \leq 1$ is the second largest singular value of the limit of $K_j$.*

For a formal statement and the analysis, see supplement Section D and Theorem 5. The result suggests that for a desirable limit being a permutation matrix, whose $\sigma_2 = 1$, the error $O(\sigma_2^{2j})$ does not drop. In practice, to expect an approximate permutation matrix, $\sigma_2 \approx 1$ and the convergence is exceedingly slow. The practical usefulness of (7) depends on the learned quality of $K_0$.

The soft and hard alignment approaches have pros and cons. The hard approach maintains the correspondence of each feature dimension of the latent vectors while the soft approach does not. Maintaining the dimension correspondence is an advantage, especially for local models that produce disentangled latent representations (Higgins et al., 2018), because each feature dimension is equipped with a semantic meaning that controls a certain aspect of the data. On the other hand, the soft approach is more straightforward and the hard approach is based on an algorithm that barely converges. In practice, we observe that neither approach decisively outperforms the other in federated inference.

## 5 LEARNING A CONSENSUS GRAPH

The example global model (4) performs a naive averaging for the local representations. Since data owners are often interconnected, a more expressive model exploits their relational interactions to improve inference (Battaglia et al., 2018). To this end, we propose to use a graph neural network (GNN) (Zhang et al., 2020; Wu et al., 2021) to process the latent representations.

Many GNNs are applicable; we focus on GCN (Kipf & Welling, 2017) for its simplicity. Let $A$ be the graph adjacency matrix and let $H$ be the matrix of aligned local representations:

$$H = \begin{bmatrix} -(P^1 h^1)^T- \\ \vdots \\ -(P^n h^n)^T- \end{bmatrix}.$$

Traditionally, GCN was designed for node classification, but we modify it slightly for our purpose as

$$\widehat{y} = \operatorname{softmax}\left( \tfrac{1}{n} \mathbf{1}^T \widehat{A} \cdot \operatorname{ReLU}(\widehat{A} H W_0) \cdot W_1 \right), \tag{8}$$

where $\widehat{A}$ is a normalization of $A$ (see (Kipf & Welling, 2017) for details) and $W_0$ and $W_1$ are weight matrices. The modification is the inclusion of $\frac{1}{n}\mathbf{1}^T$ as pooling before output. Modulo this modification, the formula (8) is a standard one used in the literature, with the bias terms omitted. It is interesting to note the equivalence of GCN (8) and the graph-agnostic model (4) when $\widehat{A}$ is replaced by the identity matrix (omitting bias terms).

In GCN, $A$ corresponds to the consensus graph among local owners as graph nodes. If such a graph is not present, it is possible to learn one such that (8) still outperforms (4). In this case, we treat $A$ as a random variable of the matrix Bernoulli distribution, where the success probabilities are free parameters to learn. Formally, the elements $A_{ij}$ are independent and each follows $\mathrm{Ber}(\theta_{ij})$, where $\theta_{ij}$ denotes the corresponding probability (Kipf et al., 2018; Shang et al., 2021). Then, the global model $g$ has $W_0$, $W_1$, the $P^i$'s, as well as $\theta$, as parameters. Following Franceschi et al. (2019); Shang et al. (2021), we formulate the training loss as an expectation over $A$'s distribution and draws a sample $A$ to obtain an unbiased estimate of the loss as well as the gradient, in each stochastic optimization step.

The central challenge of this approach is that $A$ (and hence also the loss) is not differentiable with respect to $\theta$. A popular remedy is the Gumbel softmax reparameterization trick (Jang et al., 2017; Maddison et al., 2017). In what follows, for simplicity of exposition, we treat $\theta$ a scalar rather than a matrix. The Gumbel trick works in the following manner. Let $\mathrm{Cat}(\pi)$ be the categorical distribution with probability vector $\pi$ and let $g$, of the same shape as $\pi$, be a vector variable whose elements are iid $\sim \mathrm{Gumbel}(0,1)$. Then, the vector random variable

$$y = \mathrm{softmax}((\log \pi + g)/\tau), \quad \tau > 0 \qquad (9)$$

admits a distribution converging to $\mathrm{Cat}(\pi)$ when $\tau \to 0$. Hence, to sample $\mathrm{Ber}(\theta)$ approximately but differentiably, it suffices to let $\pi = [\theta, 1 - \theta]$ and use $y_1$ as the sample.

In order to obtain one Bernoulli sample, the Gumbel trick requires to sample the Gumbel distribution twice. We consider an alternative that samples any appropriate distribution only once.

**Definition 1.** Let $F$ be the cdf of an arbitrary continuous probability distribution. Sample $s$ from this distribution and let

$$z = \mathrm{sigmoid}((F^{-1}(\theta) - s)/\tau), \quad \tau > 0. \qquad (10)$$

We call this method **icdf**.

The name icdf is owing to the use of $F^{-1}$. The reader should not confuse this method with the inverse transform method for sampling a random variable with a particular cdf $F$. Here, we use any $F$ to sample an (approximate) Bernoulli distribution. The following result qualifies $z$ to be an approximate Bernoulli variable. The proof, as well as those of subsequent theorems, is given in the supplement.

**Theorem 2.** *For all $\tau > 0$, $\theta \in (0,1)$, and $t \in [0,1]$, if the distribution with cdf $F$ is finitely supported on $[a, b]$, then*

$$\Pr(z \leq t) = \begin{cases} 0 & \text{if} \quad t < \mathrm{sigmoid}((F^{-1}(\theta) - b)/\tau), \\ 1 & \text{if} \quad t > \mathrm{sigmoid}((F^{-1}(\theta) - a)/\tau), \\ 1 - F(F^{-1}(\theta) + \tau \log(t^{-1} - 1)) & \text{otherwise.} \end{cases} \qquad (11)$$

*On the other hand, if the distribution is not finitely supported (i.e., $a = -\infty$ and/or $b = +\infty$), (11) still holds because either, or both, of the first two cases will not be invoked. As a consequence, the distribution of $z$ converges to $\mathrm{Ber}(\theta)$ as $\tau \to 0$.*

It is imperative to understand the rate of convergence of $y_1$ (Gumbel trick) and that of $z$ (icdf method). While one may take the usual convergence-in-distribution approach, the complex forms of the cdf (e.g., (11)) render the analysis difficult. Instead, we take the convergence-in-mean approach and calculate $\mathrm{Bias}(x) = \mathbb{E}[x] - \theta$. We derive the following result.

**Theorem 3.** *When $\tau$ is small,*

$$\mathrm{Bias}(y_1) = \tfrac{1}{6}\tau^2 \pi^2 \theta(1 - \theta)(1 - 2\theta) + O(\tau^4), \qquad (12)$$

$$\mathrm{Bias}(z) = \tfrac{1}{6}\tau^2 \pi^2 F''(F^{-1}(\theta)) + O(\tau^4). \qquad (13)$$

*Moreover, when $F$ is the cdf of a normal variable $\sim \mathcal{N}(0, \sigma^2)$, then*

$$\mathrm{Bias}(z) = -\tfrac{1}{6\sigma^2}\tau^2 \pi^{\frac{3}{2}} \mathrm{erf}^{-1}(2\theta - 1)e^{-(\mathrm{erf}^{-1}(2\theta-1))^2} + O(\tau^4). \qquad (14)$$

Theorem 3 suggests that the icdf method converges equally fast as does the Gumbel trick (both on the order of $O(\tau^2)$). On the other hand, the biases depend on $\theta$. Thus, one cannot set temperatures $\tau$, independently of the desired probability $\theta$, to equate the two biases. In practice, $\tau$ is a tunable hyperparameter and we use the same tuning range to fairly compare the Gumbel trick and icdf. The rationale is justified in supplement Section F.

We conclude this section by stressing the advantage of the proposed icdf method for differentiably sampling graphs for global model training: it requires fewer random number generations than does the Gumbel trick, saving time and memory.

## 6 EXPERIMENTS

In this section, we demonstrate comprehensive experiments to show that federated inference can be effectively conducted by using the proposed framework.

**Data sets.** We use four real-life, time series data sets. Two are PMU data collected from multiple data owners of the U.S. power grid. For proof of concept, we smooth out heterogeneity and prepare homogeneous data sets. Such a preprocessing is sufficient to test the proposed techniques under minimal impact of the complication by the otherwise diverse local models. Since the PMU data sets are proprietary, we also use two public, traffic data sets (Li et al., 2018) for experimentation. A summary of these data sets is given in Table 1 and the processing details are given in the supplement.

Table 1: Data sets.

|  | METR–LA | PEMS–BAY | PMU–B | PMU–C |
|---|---|---|---|---|
| # Data samples | 2856 | 4343 | 4853 | 1884 |
| # Data owners | 207 | 325 | 43 | 188 |
| Series length | 12 | 12 | 30 | 30 |
| # Features | 1 | 1 | 2 | 2 |
| # Classes | 2 | 2 | 4 | 4 |
| Missing data? | no | no | yes | yes |
| Given graph? | yes | yes | no | no |

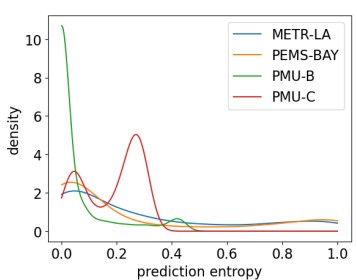

Figure 3: Distributions of prediction entropy across local models.

**Experiment setting.** All local models are LSTM (Hochreiter & Schmidhuber, 1997) with the same hyperparameters, but pretrained separately by using local data. The local models are not fine-tuned in the training of the global model. Each data set is split randomly for training/validation/testing. See the supplement for further details.

**Conflicting local predictions.** We first show that local models do not produce consistent predictions, which justifies the effort of training a global model and performing federated inference. For each datum, we compute the entropy of the predicted labels and summarize the entropies for all data into a distribution, plotted in Figure 3. Recall that the lower the entropy, the more consistent the local predictions. The figure, however, shows that a substantial amount of entropies is away from zero, suggesting that local predictions are inconsistent.

**Effectiveness of the proposed framework.** We make two sets of comprehensive comparisons to evaluate the effectiveness of the proposed framework. The first set, as outlined in Table 2, compares it with a number of straightforward baselines (A–F) and methods outside the federated inference setting (A and K). This set contains several alignment strategies for local models: (G) no alignment; (I) soft alignment; and (K) hard alignment. A straightforward variant between G and I is H, where a common weight matrix $W$ is used for all local models, serving as an alternative to alignment. Methods G to J use graph structure learning (icdf method) as the global model.

One sees that methods A to D, either lacking a local model or a global model, perform poorly as expected. Methods E to H perform better than A to D, but they lack a proper alignment of the local models and hence are outperformed by methods I and J that perform alignment. Between the two strategies, neither decisively wins over the other. The advantage of soft alignment is its simplicity and that of hard alignment is the preservation of neuron correspondence. Finally, method K (end-to-

Table 2: Effectiveness of latent alignment in a graph-based global model. Superscript numbers are standard deviations. * Note that A and K are not applicable to the federated inference setting.

| | METR-LA | | PEMS-BAY | | PMU-B | | PMU-C | |
|---|---|---|---|---|---|---|---|---|
| | F1 | AUC | F1 | AUC | F1 | AUC | F1 | AUC |
| A: Common model * | $.255^{.000}$ | - | $.334^{.000}$ | - | $.360^{.000}$ | - | $.286^{.000}$ | - |
| B: Local model + majority voting | $.114^{.000}$ | - | $.089^{.000}$ | - | $.291^{.000}$ | - | $.182^{.000}$ | - |
| C: Local model + binary threshold | $.692^{.000}$ | - | $.639^{.000}$ | - | - | - | - | - |
| D: Best local model | $.528^{.000}$ | $.702^{.000}$ | $.553^{.000}$ | $.792^{.000}$ | $.370^{.000}$ | $.692^{.000}$ | $.324^{.000}$ | $.618^{.000}$ |
| E: Local model + MLP (Eqn (4)) | $.768^{.009}$ | $.957^{.004}$ | $.738^{.012}$ | $.935^{.001}$ | $.391^{.003}$ | $.727^{.006}$ | $.342^{.008}$ | $.636^{.010}$ |
| F: Local model + concatenation | $.824^{.006}$ | $.971^{.001}$ | $.854^{.003}$ | $.979^{.002}$ | $.386^{.005}$ | $.693^{.064}$ | $.389^{.018}$ | $.698^{.010}$ |
| G: Local model + icdf (no align.) | $.798^{.009}$ | $.963^{.004}$ | $.755^{.009}$ | $.943^{.001}$ | $.387^{.003}$ | $.734^{.015}$ | $.380^{.006}$ | $.658^{.005}$ |
| H: Local model + shared $W$ + icdf | $.817^{.009}$ | $.966^{.001}$ | $.747^{.009}$ | $.941^{.004}$ | $.387^{.006}$ | $.725^{.010}$ | $.368^{.012}$ | $.660^{.008}$ |
| I: Local model + soft align. + icdf | $.835^{.010}$ | $.975^{.001}$ | $.860^{.005}$ | $.980^{.002}$ | $.390^{.008}$ | $.734^{.008}$ | $.444^{.027}$ | $.693^{.011}$ |
| J: Local model + hard align. + icdf | $.839^{.006}$ | $.973^{.001}$ | $.855^{.008}$ | $.976^{.001}$ | $.390^{.004}$ | $.737^{.016}$ | $.404^{.016}$ | $.686^{.008}$ |
| K: J + end-to-end * | $.825^{.012}$ | $.973^{.002}$ | $.823^{.006}$ | $.972^{.002}$ | $.382^{.007}$ | $.717^{.010}$ | $.392^{.020}$ | $.683^{.006}$ |

Table 3: Comparison of global models. * Some references of rows are with respect to Table 2.

| | | METR-LA | | PEMS-BAY | | PMU-B | | PMU-C | |
|---|---|---|---|---|---|---|---|---|---|
| | | F1 | AUC | F1 | AUC | F1 | AUC | F1 | AUC |
| No align | No graph | $.768^{.009}$ | $.957^{.004}$ | $.738^{.012}$ | $.935^{.001}$ | $.391^{.003}$ | $.727^{.006}$ | $.342^{.008}$ | $.636^{.010}$ |
| | Given graph | $.763^{.020}$ | $.957^{.007}$ | $.742^{.024}$ | $.942^{.005}$ | - | - | - | - |
| | Gumbel | $.785^{.008}$ | $.959^{.006}$ | $.751^{.008}$ | $.942^{.001}$ | $.387^{.001}$ | $.730^{.017}$ | $.381^{.025}$ | $.658^{.006}$ |
| | icdf (row G) * | $.798^{.009}$ | $.963^{.004}$ | $.755^{.009}$ | $.943^{.001}$ | $.387^{.003}$ | $.734^{.015}$ | $.380^{.006}$ | $.658^{.005}$ |
| Soft align | No graph | $.833^{.010}$ | $.975^{.001}$ | $.846^{.008}$ | $.977^{.001}$ | $.388^{.001}$ | $.736^{.015}$ | $.386^{.008}$ | $.694^{.005}$ |
| | Given graph | $.828^{.007}$ | $.974^{.001}$ | $.854^{.003}$ | $.977^{.001}$ | - | - | - | - |
| | Gumbel | $.834^{.016}$ | $.975^{.001}$ | $.863^{.014}$ | $.980^{.001}$ | $.390^{.003}$ | $.733^{.012}$ | $.435^{.028}$ | $.693^{.007}$ |
| | icdf (row I) * | $.835^{.010}$ | $.975^{.001}$ | $.860^{.005}$ | $.980^{.002}$ | $.390^{.008}$ | $.734^{.008}$ | $.444^{.027}$ | $.693^{.011}$ |
| Hard align | No graph | $.825^{.008}$ | $.971^{.003}$ | $.847^{.008}$ | $.976^{.001}$ | $.387^{.004}$ | $.736^{.007}$ | $.372^{.004}$ | $.674^{.015}$ |
| | Given graph | $.829^{.014}$ | $.971^{.002}$ | $.848^{.010}$ | $.973^{.002}$ | - | - | - | - |
| | Gumbel | $.837^{.013}$ | $.973^{.002}$ | $.850^{.009}$ | $.976^{.001}$ | $.391^{.004}$ | $.732^{.014}$ | $.410^{.016}$ | $.687^{.004}$ |
| | icdf (row J) * | $.839^{.006}$ | $.973^{.001}$ | $.855^{.008}$ | $.976^{.001}$ | $.390^{.004}$ | $.737^{.016}$ | $.404^{.016}$ | $.686^{.008}$ |

end training) performs worse than method J (separate training). The result is not surprising, because joint training compromises the optimality of local data representations separately obtained by each local model. We also note this method is outside the setting of federated inference and generally cannot be used unless data owners agree to share data.

**Comparison of global models.** The other set of comparisons, as outlined in Table 3, extends each alignment strategy (including no alignment) to the role of graphs in the global model: not using a graph, using the given graph, and learning a graph (by using either the Gumbel trick or the icdf method). The numbers in the table suggest that within each alignment strategy, graph structure learning significantly improves the classification. The performance of the Gumbel trick and that of icdf is highly comparable.

**Quality of learned permutations.** For hard alignment, we investigate the learning of the permutation matrices. According to Theorem 1, $\sigma_2^2$ of the limit of $K_j$ dictates the convergence speed. Since we do not know the limit, we compute $\sigma_2^2$ of $K_T$ and summarize them in Table 4 for some local model in each data set, under several involved methods. One sees that all values are close to 1, suggesting that the convergence is indeed rather slow, agreeing with theory. Note that some values are greater than 1 because $K_T$ is not strictly doubly stochastic (owing to slow convergence).

In Figure 4, we visualize $K_T$ for some local model in each data set. The plots clearly show patterns of a permutation matrix: there is one and only one significant value per row and per column. Because of the slow convergence, we attribute the desirable results of $K_T$ (at a small $T$) to the success of the learning of $K_0$. Note also interestingly that a learned permutation may be the identity mapping.

Table 4: Examples of squared second singular value, $\sigma_2^2$, of $K_T$.

| | | METR-LA | PEMS-BAY | PMU-B | PMU-C |
|---|---|---|---|---|---|
| **Hard align** | No graph | 1.007 | 1.000 | 1.000 | 1.000 |
| | Given graph | 1.010 | 1.000 | - | - |
| | Gumbel | 1.007 | 1.002 | 1.001 | 1.019 |
| | icdf | 1.000 | 1.000 | 1.000 | 1.025 |

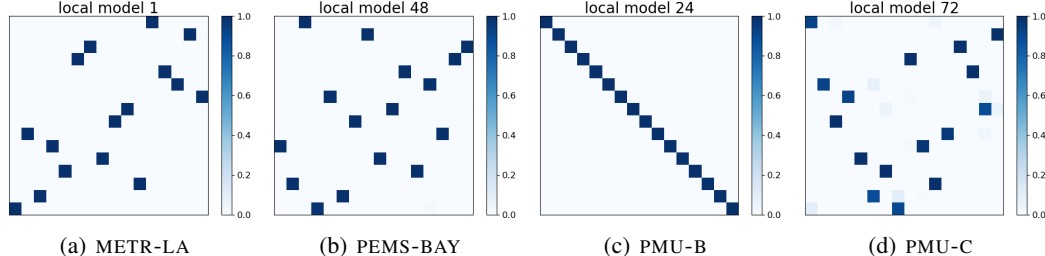

(a) METR-LA  (b) PEMS-BAY  (c) PMU-B  (d) PMU-C

Figure 4: Examples of learned permutation matrices ($K_T$). All are from Method H.

**Comparison of Gumbel softmax and icdf.** Prior results suggest that these two approaches for differentiably sampling the Bernoulli distribution perform equally well. An advantage of icdf is its lower computational cost. To demonstrate this advantage, we design a mini-benchmark that highlights the sampling and gradient computation and minimizes the effect of irrelevant complications (such as permutation and GNN). To this end, we generate samples $(x_i \in \mathbb{R}^n, y_i \in \mathbb{R}^n)$ for some $A \in \{0,1\}^{n \times n}$, where $y_i = Ax_i + \text{noise}$, and use the samples to learn $A$ through differentiable parameterization. Figure 5 shows the time and memory consumption at a fixed number of learning epochs. As a sanity check, the running time scales nicely as $O(n^2)$ as expected (while the memory consumption is complicated; it does not follow $O(n^2)$ because of memory management in Python). Overall, one clearly sees the lower computational cost of the icdf method.

We also report the time and memory consumption for the experiments on the four data sets; see Table 5 in the supplement. The results well agree that the icdf method is more economic.

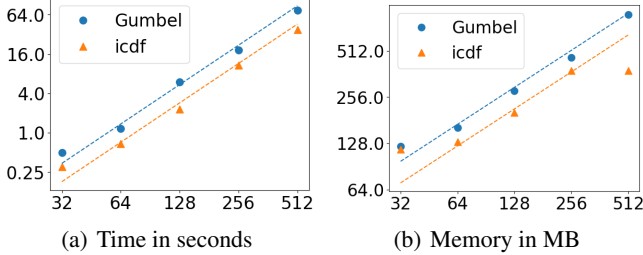

(a) Time in seconds  (b) Memory in MB

Figure 5: Time and memory consumption as the matrix size ($n$, horizontal axis) increases.

## 7 CONCLUSIONS

In this paper, we study federated inference, a less addressed scenario of machine learning with distributed data that require collective inference. This scenario is in contrast to federated learning, where inference is local and requires no joint efforts. We motivate the practicality of federated inference by using a power grid example and propose a local–global model framework for it. Two important components of the framework are the alignment of the data representations produced by local models and the learning of the global model by using a graph neural network. Comprehensive experiments suggest the feasibility of federated inference and the effectiveness of the framework.

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

## A    RELATED WORK

The concept of federated learning was first coined by McMahan et al. (2017) and it has attracted surging interests since. A form of distributed optimization, federated learning is faced with data challenges beyond conventional assumptions and puts communication efficiency and data privacy as primary concerns. Recent surveys (Yang et al., 2019; Li et al., 2019; Kairouz et al., 2019) comprehensively study the subject, review systems and infrastructures, and suggest open problems.

The typical setting of federated learning is that data sets across owners share the same feature space but differ in samples. Besides this horizontal partitioning of the data matrix, a vertical partitioning was studied by Hardy et al. (2017); Nock et al. (2018); Heinze et al. (2014; 2016), wherein features are split across owners instead. This setting bears resemblance to our federated inference scenario, but a crucial distinction is that existing methods for vertical federated learning all perform joint training. In the referenced work, to preserve privacy, encrypted data or randomly projected data are communicated among data owners as well as a central coordinator. Such an approach incurs demanding communication for many owners. Recently, Chen et al. (2020) study a different model, whose parameters are distributed among owners as well as a central server. The part of the model corresponding to an owner bears resemblance to our local models; but they are not local models since they are not independently trained by using local data. Another work along a similar direction is conducted by Hu et al. (2019), but the global model has no parameters; it is merely a sum of the local outputs followed by activation (e.g., sigmoid for classification).

Our framework learns parameter matrices to align local representations. Such alignments similarly appear in model fusion, where a number of models are fused together into a common model through aligning model parameters (Yurochkin et al., 2019a). In the context of deep learning, if the neural networks come from the same model family, their weights can be matched layerwise, even if the numbers of weights are different (Yurochkin et al., 2019b; Wang et al., 2020). The referenced work treats the problem as a bipartite graph matching, where the cost matrix is inferred from maximum a posteriori estimation. Then, the Hungarian algorithm (Kuhn, 1955) is applied to find the matching. In our work, instead we treat the permutation alignment as a differentiable parameterization with the help of Sinkhorn–Knopp (Sinkhorn & Knopp, 1967; Mena et al., 2018; Emami & Ranka, 2018), so that it can be learned end-to-end with other parameters of the global model.

Our framework also advocates learning a graph of data owners in the global model. Graph structure learning appears under various contexts. One field of study is probabilistic graphical models and casual inference, whereby a directed acyclic structure is learned. Gradient-based approaches in this context include Zheng et al. (2018); Yu et al. (2019); Lachapelle et al. (2020). On the other hand, a general graph may still be useful without resorting to causality. Recent approaches supporting GNN-based modeling include Kipf et al. (2018); Franceschi et al. (2019); Wu et al. (2020); Shang et al. (2021), wherein a graph structure is simultaneously learned together with the GNN parameters. The Gumbel trick (Jang et al., 2017; Maddison et al., 2017) is frequently used for differentiable parameterization, but in this paper we study a more economic alternative parameterization, icdf.

## B    PERMUTATION AMBIGUITY EXAMPLE FOR GRU

In section 3, we discuss that one can arbitrarily permute the latent representations while keeping a local model fixed. Here, we give another example—the GRU. Let $x = \{x_1, x_2, \ldots, x_T\}$ be an input sequence. The embedding function $h = \text{embedding}(x)$ implemented as a GRU reads:

```
 1: function h = GRU({x_t}_{t=1,...,T})
 2:     h_0 = 0
 3:     for t = 1, ..., T do
 4:         z_t = sigmoid(W_z x_t + U_z h_{t-1} + b_z)
 5:         r_t = sigmoid(W_r x_t + U_r h_{t-1} + b_r)
 6:         n_t = tanh(W_n x_t + U_n(r_t ⊙ h_{t-1}) + b_n)
 7:         h_t = (1 - z_t) ⊙ h_{t-1} + z_t ⊙ n_t
 8:     end for
 9:     return h = h_T
10: end function
```

One can arbitrarily permute the elements of $h$ through manipulating the GRU parameters properly. To achieve $h[\mathrm{p}] = \mathrm{embedding}(x; \mathrm{p})$,

- the gate outputs and bias vectors ($z_t$, $r_t$, $n_t$, $h_t$, $b_z$, $b_r$, $b_n$) will need be permuted accordingly ($z_t[\mathrm{p}]$, $r_t[\mathrm{p}]$, $n_t[\mathrm{p}]$, $h_t[\mathrm{p}]$, $b_z[\mathrm{p}]$, $b_r[\mathrm{p}]$, $b_n[\mathrm{p}]$);

- the weight matrices attached to the input ($W_z$, $W_r$, $W_n$) will need to have their rows (i.e., output neurons) permuted ($W_z[\mathrm{p}, :]$, $W_r[\mathrm{p}, :]$, $W_n[\mathrm{p}, :]$); and

- the weight matrices attached to the hidden states ($U_z$, $U_r$, $U_n$) will need to have both their rows and columns permuted ($U_z[\mathrm{p}, \mathrm{p}]$, $U_r[\mathrm{p}, \mathrm{p}]$, $U_n[\mathrm{p}, \mathrm{p}]$).

## C  FEDERATED INFERENCE FRAMEWORK

The framework is summarized in Algorithm 1. Details are elaborated in Sections 4 and 5.

---

**Algorithm 1** Federated Inference Framework

---

1: **function** TRAINING($\{(x^1)_j, (x^2)_j, \ldots, (x^n)_j, y_j\}_{j=1,\ldots,N}$)
2:     Each data owner trains a local model $f^i$ with its local data part $\{(x^i)_j\}_j$ and label $\{y_j\}_j$
3:     Each data owner sends all local data representations $\{(h^i)_j\}_j$ to the central server
4:     Central server trains a global model $\widehat{y} = g(P^1 h^1, P^2 h^2, \ldots, P^n h^n)$ by using $\{(h^i)_j\}_j$ gathered from data owners and labels $\{y_j\}_j$. Here,
   - The global model is (8), where the loss is taken over the distribution of $\widehat{A}$, whose entries are sampled by using (10);
   - Each alignment matrix $P^i$ is either an arbitrary parameter matrix (soft alignment), or a doubly stochastic matrix $K_T$ computed by (7) using $T$ steps (hard alignment).
5: **end function**

6: **function** INFERENCE($x^1, \ldots, x^n$)
7:     Each data owner evaluates its local model with $x^i$ to obtain $h^i$ and sends to the central server
8:     Central server evaluates the global model by taking all $h^i$ as input and produces prediction
9: **end function**

---

## D  FURTHER DETAILS OF SECTION 4 (PARAMETERIZATION OF PERMUTATION)

The permutation matrix $P$ is a combinatorial object and is unsuitable for gradient-based training. We relax $P$ by a doubly stochastic matrix, which is nonnegative and whose row sums and column sums are all $1$. A benefit of such a relaxation is that a differentiable computational procedure exists for doubly stochastic matrices. The procedure, called the Sinkhorn–Knopp algorithm (Sinkhorn & Knopp, 1967), was recently used by Mena et al. (2018) and Emami & Ranka (2018) in other contexts. Informally speaking, starting from a nonnegative square matrix $K_0$, repeatedly normalize its rows and columns by their sums until convergence. The limit is a doubly stochastic matrix.

In fact, Mena et al. (2018) and Emami & Ranka (2018) do not start from a free $K_0$ but parameterize it as $\exp(M/\tau)$, where $M$ is a free parameter matrix and $\tau > 0$ is a hyperparameter. Such a parameterization follows more closely Sinkhorn's algorithm (Sinkhorn, 1964) for computing the optimal transport (Peyré & Cuturi, 2019) under an entropic regularization (Wilson, 1969), where $\tau$ is supposed to anneal to zero. However, this parameterization is not imperative and is not the only option. For simplicity of exposition, we treat $K_0$ as a free parameter.

To understand the convergence of this method, we rewrite the algorithm equivalently as

$$\text{Start with } r_0 = \mathbf{1}, \text{ for } j = 0, 1, \ldots, \quad c_{j+1} = \mathbf{1} \oslash (K_0^T r_j) \text{ and } r_{j+1} = \mathbf{1} \oslash (K_0 c_{j+1}). \quad (15)$$

Here, the $c_j$'s and $r_j$'s correspond to the column sums and row sums of the iterating matrix in the above informal description. Let $\mathrm{diag}(x)$ be a diagonal matrix whose diagonal is $x$. The following result is proved by Sinkhorn & Knopp (1967).

**Theorem 4** (Sinkhorn–Knopp). *If $K_0$ is nonnegative, then a necessary and sufficient condition that the iteration* (15) *converges is that $K_0$ has total support. Let positive vectors $c_*$ and $r_*$ be the limits of $\{c_j\}$ and $\{r_j\}$, respectively. The matrix $\mathrm{diag}(r_*)K_0 \mathrm{diag}(c_*)$ is doubly stochastic.*

This result, however, does not give the convergence speed. In order to obtain a tight one, we follow Knight (2008) and perform a minor modification of (15):

Start with $r_0 = c_0 = \mathbf{1}$, for $j = 0, 1, \ldots, \quad c_{j+1} = \mathbf{1} \oslash (K_0^T r_j)$ and $r_{j+1} = \mathbf{1} \oslash (K_0 c_j)$. $\qquad$ (7)

This modification changes nothing essential of the algorithm, since it is equivalent to applying (15) on $\begin{bmatrix} 0 & K_0 \\ K_0^T & 0 \end{bmatrix}$. We define $K_j = \mathrm{diag}(r_j)K_0 \mathrm{diag}(c_j)$ and derive the following result. Its proof is given in Section D.1.

**Theorem 5.** *If $K_0$ is nonnegative and irreducible, then the iteration* (7) *will converge linearly. Let positive vectors $c_*$ and $r_*$ be the limits of $\{c_j\}$ and $\{r_j\}$, respectively. Then, $K_* = \mathrm{diag}(r_*)K_0 \mathrm{diag}(c_*)$ is the limit of $K_j$ and is doubly stochastic. Furthermore, there exist a positive constant $C$ and a positive integer $J$ such that for all $j \geq J$,*

$$\left\| \begin{bmatrix} K_j^T \mathbf{1} \\ K_j \mathbf{1} \end{bmatrix} - \begin{bmatrix} \mathbf{1} \\ \mathbf{1} \end{bmatrix} \right\| \leq C(1 + \sigma_2^2)\sigma_2^{2(j-J)}T_J \quad with \quad T_j := \left\| \begin{bmatrix} c_j \\ r_j \end{bmatrix} - \begin{bmatrix} c_* \\ r_* \end{bmatrix} \right\|, \qquad (16)$$

*where $\sigma_2 < 1$ is the second largest singular value of $K_*$.*

A few important points of Theorem 5 are noted. First, the convergence (16) says that the row sums and column sums of $K_j$ approach 1 at a speed $O(\sigma_2^{2j})$. Second, the iteration (7) preserves the irreducibility of the $K_j$'s and thus of the limit $K_*$. Then, by the Perron–Frobenius Theorem (Horn & Johnson, 2012), the doubly stochastic matrix $K_*$ has a *simple* eigenvalue 1, which coincides with the spectral radius as well as the dominant singular value. Thus, the second singular value, $\sigma_2$, must be $< 1$. The farther away it is from 1, the faster the iteration converges.

Unfortunately, Theorem 5 is not applicable to a permutation matrix, because such a matrix is not irreducible. However, the theorem is practically useful under perturbation theory. After all, the limit $K_*$ is hardly an exact permutation matrix; at best, it is $\epsilon$-close. Note that all singular values of a permutation matrix $P$ is 1. If $K_*$ is different from $P$ by $\epsilon$ in the spectral norm, then according to Weyl's Theorem (Horn & Johnson, 2012), $\sigma_2 \geq 1 - \epsilon$.

The above analysis indicates that the Sinkhorn–Knopp algorithm (7) will be extremely slow to converge if $\epsilon$ is small. A natural question is, is this parameterization useful? We argue positively. In practice, we run the procedure (7) with only a few steps, $T$. The iterate $K_T$ can still be reasonably close to a permutation matrix, when the starting point $K_0$ is sufficiently good. In essence, we rely on the training of the global model to find a good starting point and on the procedure (7) to normalize its values to $[0, 1]$. One may further use an entropy regularization

$$\sum_{\ell} \mathrm{entropy}(K_T[:, \ell]) + \mathrm{entropy}(K_T[\ell, :])$$

to push the values toward the two extremes, 0 or 1.

### D.1 PROOF OF THEOREM 5

Applying the iteration formula (7), it is straightforward to see that

$$K_j^T \mathbf{1} - \mathbf{1} = (c_j \oslash c_{j+1}) - \mathbf{1} \quad and \quad K_j \mathbf{1} - \mathbf{1} = (r_j \oslash r_{j+1}) - \mathbf{1}.$$

We expand the left equation to

$$(c_j \oslash c_{j+1}) - \mathbf{1} = (c_j - c_* + c_* - c_{j+1}) \oslash c_{j+1}$$

and similarly for the right equation. Then,

$$\begin{bmatrix} K_j^T \mathbf{1} - \mathbf{1} \\ K_j \mathbf{1} - \mathbf{1} \end{bmatrix} = \left( \begin{bmatrix} c_j - c_* \\ r_j - r_* \end{bmatrix} + \begin{bmatrix} c_* - c_{j+1} \\ r_* - r_{j+1} \end{bmatrix} \right) \oslash \begin{bmatrix} c_{j+1} \\ r_{j+1} \end{bmatrix}.$$

Applying the triangle inequality and the property $\|x \oslash y\| \leq \|x\| / \min |y_i|$, we obtain

$$\left\| \begin{bmatrix} K_j^T \mathbf{1} - \mathbf{1} \\ K_j \mathbf{1} - \mathbf{1} \end{bmatrix} \right\| \leq \left( \left\| \begin{bmatrix} c_j - c_* \\ r_j - r_* \end{bmatrix} \right\| + \left\| \begin{bmatrix} c_{j+1} - c_* \\ r_{j+1} - r_* \end{bmatrix} \right\| \right) / \tau_{j+1}, \qquad (17)$$

where $\tau_{j+1}$ denotes the absolute minimum of the elements of $c_{j+1}$ and $r_{j+1}$.

We will need the following result from Knight (2008).

**Theorem 6.** *If $K_0$ is nonnegative and irreducible, then there exists a positive integer $J$ such that for all $j \geq J$,*

$$\left\| \begin{bmatrix} c_{j+1} \\ r_{j+1} \end{bmatrix} - \begin{bmatrix} c_* \\ r_* \end{bmatrix} \right\| \leq \sigma_2^2 \left\| \begin{bmatrix} c_j \\ r_j \end{bmatrix} - \begin{bmatrix} c_* \\ r_* \end{bmatrix} \right\|.$$

Theorem 6 states that $T_{j+1} \leq \sigma_2^2 T_j$ for all $j \geq J$. Because $\sigma_2 < 1$, $T_j$ is monotonically decreasing once $j \geq J$. Moreover, because $c_j$ and $r_j$ are positive vectors for all $j$, when $j \geq J$, the elements of $c_j$ and $r_j$ are bounded away from zero. Let $\tau > 0$ be the infimum of the elements of all $c_j$ and $r_j$ when $j > J$; (17) becomes

$$\left\| \begin{bmatrix} K_j^T \mathbf{1} - \mathbf{1} \\ K_j \mathbf{1} - \mathbf{1} \end{bmatrix} \right\| \leq \frac{T_j + T_{j+1}}{\tau}.$$

Thus, applying the relation $T_{j+1} \leq \sigma_2^2 T_j$, we obtain

$$\left\| \begin{bmatrix} K_j^T \mathbf{1} - \mathbf{1} \\ K_j \mathbf{1} - \mathbf{1} \end{bmatrix} \right\| \leq \frac{(1 + \sigma_2^2) T_j}{\tau} \leq \frac{(1 + \sigma_2^2) \sigma_2^{2(j-J)} T_J}{\tau},$$

which concludes the proof by taking $C = 1/\tau$.

# E PROOFS AND ADDITIONAL RESULTS OF SECTION 5 (BERNOULLI SAMPLING)

## E.1 DISTRIBUTION OF GUMBEL SOFTMAX

As preliminary, we consider the first entry $y_1$ of the random variable $y$ defined in (9) for the Gumbel softmax parameterization. Note that for any $\tau \neq 0$, $y_1$ is only approximately binary; the possible values of $y_1$ in fact span the entire interval $[0, 1]$. We derive the following cdf for $y_1$. Recall that for notational simplicity, $\theta$ denotes a scalar rather than a matrix.

**Theorem 7.** *For all $\tau > 0$, $\theta \in (0, 1)$, and $t \in [0, 1]$, we have*

$$\Pr(y_1 \leq t) = \frac{t^\tau (1 - \theta)}{t^\tau (1 - \theta) + (1 - t)^\tau \theta}. \tag{18}$$

**Proof.** We first consider the case $0 < t < 1$. Through simple algebraic manipulation, we obtain that $y_1 \leq t$ is equivalent to

$$g_1 - g_2 \leq \tau \log \frac{t}{1-t} - \log \frac{\theta}{1-\theta}. \tag{19}$$

Let $g_1 = -\log(-\log u)$ and $g_2 = -\log(-\log v)$, where $u$ and $v$ are independent and $\sim \mathcal{U}(0, 1)$. Then, (19) is equivalent to

$$v \geq u^M \quad \text{where} \quad M = \frac{t^\tau (1 - \theta)}{(1 - t)^\tau \theta}.$$

Therefore, by recalling that $u$ and $v$ are uniform in $[0, 1]^2$, we note that the probability that $v \leq u^M$ happens is the double integral

$$\Pr(v \geq u^M) = \int_0^1 \int_{u^M}^1 1 \, dv du.$$

This integral is nothing but

$$1 - \int_0^1 u^M \, du = \frac{M}{1 + M},$$

which completes the proof of (18).

The cases of $t = 0$ or $1$ obviously hold by continuity.

### E.2 PROOF OF THEOREM 2

We first consider the case when the distribution with cdf $F$ is finitely supported on $[a, b]$. Through simple algebraic manipulation, we obtain that $z \leq t$ is equivalent to $s \geq M$ where $M := F^{-1}(\theta) + \tau \log(t^{-1} - 1)$. If $t < \text{sigmoid}((F^{-1}(\theta) - b)/\tau)$, we see that $M > b$ and thus such $s$ can never occur. Similarly, if $t > \text{sigmoid}((F^{-1}(\theta) - a)/\tau)$, we see that $M < a$, which indicates that $s \geq M$ always happens. Otherwise, when $t$ is within the two extremes, the probability that $s \geq M$ happens is $1 - F(M)$, concluding the proof of (11).

The statement of the theorem regarding the case when the distribution is not finitely supported is obviously true.

To show that the distribution of $z$ converges to $\text{Ber}(\theta)$, let us first consider the scenario when the distribution with cdf $F$ is finitely supported. The cdf of $z$ (see (11)) is always continuous but it has three segments connected by two joints: $t_1 = \text{sigmoid}((F^{-1}(\theta) - b)/\tau)$ and $t_2 = \text{sigmoid}((F^{-1}(\theta) - a)/\tau)$. When $\tau \to 0$, the joint $t_1 \to 0$ and the joint $t_2 \to 1$ and thus the middle segment has a wider and wider support converging to $[0, 1]$. Hence, it suffices to consider only the middle segment. Further, with an analogous argument for other scenarios, it is also true that it suffices to consider only the third case of (11).

In this case, for any fixed $t < 1$ and when $\tau \to 0$, we have $\tau \log(t^{-1} - 1) \to 0$ and thus $\Pr(z \leq t) \to 1 - F(F^{-1}(\theta)) = 1 - \theta$. Meanwhile, we cannot push $\tau \to 1$ because then the limit of $\tau \log(t^{-1} - 1)$ is undefined. However, we know by definition that $\Pr(z \leq 1) = 1$. Hence, the continuous distribution of $z$ converges to a degenerate distribution $\Pr(z < 1) = 1 - \theta$ and $\Pr(z = 1) = 1$. This is the cdf of $\text{Ber}(\theta)$.

### E.3 PROOF OF THEOREM 3

By the definition of bias, we have

$$\text{Bias}(x) = \mathbb{E}[x] - \theta \quad \text{where} \quad \mathbb{E}[x] = \int_0^1 t \, d\Pr(x \leq t) = 1 - \int_0^1 \Pr(x \leq t) \, dt.$$

Therefore, for Gumbel softmax,

$$\text{Bias}(y_1) = 1 - \theta - \int_0^1 \frac{t^\tau (1 - \theta)}{t^\tau (1 - \theta) + (1 - t)^\tau \theta} \, dt,$$

and for icdf with any $F$,

$$\text{Bias}(z) = \int_0^1 F(F^{-1}(\theta) + \tau \log(t^{-1} - 1)) \, dt - \theta.$$

We now prove Theorem 3 in a few parts.

**Proof of (13).** Let $s = F^{-1}(\theta)$ and perform a change of variable $m = \log(t^{-1} - 1)$. Then,

$$\text{Bias}(z) = \int_0^1 [F(s + \tau m) - F(s)] \, dt = \int_{-\infty}^\infty [F(s + \tau m) - F(s)] \frac{e^m}{(1 + e^m)^2} \, dm.$$

We perform Taylor expansion of $F$ around $s$ and obtain

$$F(s + \tau m) - F(s) = \sum_{n=1}^\infty \frac{F^{(n)}(s)}{n!} \tau^n m^n.$$

Therefore,

$$\text{Bias}(z) = \sum_{n=1}^\infty \frac{F^{(n)}(s)}{n!} \tau^n \int_{-\infty}^\infty \frac{m^n e^m}{(1 + e^m)^2} \, dm$$

Each integral term is finite and the odd terms vanish because the integrands are odd functions. Thus, for small $\tau$, we are left with

$$\text{Bias}(z) = \frac{F''(s)}{2} \tau^2 \int_{-\infty}^\infty \frac{m^2 e^m}{(1 + e^m)^2} \, dm + O(\tau^4).$$

The definite integral evaluates to $\frac{\pi^2}{3}$; we therefore conclude the proof.

**Proof of** (14). Equation (14) is straightforward by substuting

$$F''(s) = -\frac{s}{\sigma^3 \sqrt{2\pi}} e^{-\frac{s^2}{2\sigma^2}} = -\frac{\text{erf}^{-1}(2\theta - 1)}{\sigma^2 \sqrt{\pi}} e^{-(\text{erf}^{-1}(2\theta-1))^2}.$$

into (13).

**Proof of** (12). To simplify notation, let $\beta = \theta/(1 - \theta)$ and perform a change of variable $m = \log(t^{-1} - 1)$. Then,

$$\int_0^1 \frac{t^\tau (1 - \theta)}{t^\tau (1 - \theta) + (1 - t)^\tau \theta} \, dt = \int_0^1 \frac{dt}{1 + \beta e^{m\tau}} = \int_{-\infty}^\infty \frac{1}{1 + \beta e^{m\tau}} \frac{e^m}{(1 + e^m)^2} \, dm.$$

Denote $h(\tau, m) = [1 + \beta e^{m\tau}]^{-1}$. Treating $h$ a function of $\tau$ and performing Taylor expansion around zero, we obtain

$$h(\tau, m) = \sum_{n=0}^\infty \frac{h^{(n)}(0, m)}{n!} \tau^n.$$

Therefore,

$$\int_0^1 \frac{t^\tau (1 - \theta)}{t^\tau (1 - \theta) + (1 - t)^\tau \theta} \, dt = \sum_{n=0}^\infty \frac{\tau^n}{n!} \int_{-\infty}^\infty h^{(n)}(0, m) \frac{e^m}{(1 + e^m)^2} \, dm.$$

In a moment, we will show that for all $n$,

$$h^{(n)}(0, m) = C_n m^n \quad \text{where } C_n \text{ is independent of } m. \tag{20}$$

Suppose that (20) holds. Then, each integral term is finite and the odd terms vanish, because the integrands are odd functions. Therefore, for small $\tau$, we are left with

$$\int_0^1 \frac{t^\tau (1 - \theta)}{t^\tau (1 - \theta) + (1 - t)^\tau \theta} \, dt = C_0 \int_{-\infty}^\infty \frac{e^m}{(1 + e^m)^2} \, dm + C_2 \frac{\tau^2}{2} \int_{-\infty}^\infty \frac{m^2 e^m}{(1 + e^m)^2} \, dm + O(\tau^4).$$

By calculating

$$C_0 = h(0, m) = [1 + \beta]^{-1} = 1 - \theta, \qquad C_2 = h''(0, m) = -\theta(1 - \theta)(1 - 2\theta),$$

$$\int_{-\infty}^\infty \frac{e^m}{(1 + e^m)^2} \, dm = 1, \qquad \int_{-\infty}^\infty \frac{m^2 e^m}{(1 + e^m)^2} \, dm = \frac{\pi^2}{3},$$

we conclude that

$$\text{Bias}(y_1) = \frac{\tau^2 \pi^2 \theta(1 - \theta)(1 - 2\theta)}{6} + O(\tau^4).$$

It remains to prove (20). We suppress the argument on $m$ and write $g(\tau) = 1 + \beta e^{m\tau}$ and $h(\tau) = g(\tau)^{-1}$. By Faà di Bruno's formula,

$$h^{(n)}(0) = \left( \frac{1}{g(\tau)} \right)^{(n)} \bigg|_{\tau=0} = \sum_{k=1}^n \frac{(-1)^k k!}{g(0)^{k+1}} \cdot B_{n,k}\Big(g'(0), g''(0), \ldots, g^{(n-k+1)}(0)\Big),$$

where $B_{n,k}$ is the Bell polynomial. Clearly, $g(0) = 1 + \beta$ and $g^{(r)}(0) = \beta m^r$ for all $r > 0$. Hence, $B_{n,k}$ is a multiple of $m^n$. Therefore, $h^{(n)}(0)$ is a multiple of $m^n$.

### E.4 Additional Result Regarding the Bias

Theorem 3 states results for a small temperature $\tau$. The purpose is to understand the limiting behavior of the bias. Here, we give an additional result for any $\tau > 0$. It states that the biases of the two sampling approaches have the same sign. This result is a nontrivial extension of Theorem 3 and requires a different proof technique.

**Theorem 8.** *For any $\tau > 0$,*

$$\text{Bias}(y_1) > 0 \text{ when } \theta < \tfrac{1}{2}, \quad \text{Bias}(y_1) = 0 \text{ when } \theta = \tfrac{1}{2}, \quad \text{Bias}(y_1) < 0 \text{ when } \theta > \tfrac{1}{2}. \quad (21)$$

*Moreover, if $F'(x)$ (that is, the pdf) is even and is increasing when $x < 0$, then*

$$\text{Bias}(z) > 0 \text{ when } \theta < \tfrac{1}{2}, \quad \text{Bias}(z) = 0 \text{ when } \theta = \tfrac{1}{2}, \quad \text{Bias}(z) < 0 \text{ when } \theta > \tfrac{1}{2}. \quad (22)$$

We prove Theorem 8 in two parts.

**Proof of** (21)**.** Consider

$$\text{Bias}(y_1) = \int_0^1 g(t, \theta)\, dt \quad \text{where} \quad g(t, \theta) = 1 - \theta - \frac{t^\tau(1-\theta)}{t^\tau(1-\theta) + (1-t)^\tau\theta}.$$

With a bruteforce calculation, we have

$$g(t,\theta) + g(1-t,\theta) = \frac{[(1-t)^\tau - t^\tau]^2\theta(1-\theta)(1-2\theta)}{[t^\tau(1-\theta) + (1-t)^\tau\theta][(1-t)^\tau(1-\theta) + t^\tau\theta]}.$$

All terms on the right-hand side are positive, except $1 - 2\theta$. Therefore, when $\theta < \tfrac{1}{2}$, $g(t,\theta) + g(1 - t,\theta) > 0$ and hence

$$\text{Bias}(y_1) = \int_0^1 \frac{g(t,\theta) + g(1-t,\theta)}{2}\, dt > 0.$$

The other cases ($\theta > \tfrac{1}{2}$ and $\theta = \tfrac{1}{2}$) are similarly proved.

**Proof of** (22)**.** Consider

$$\text{Bias}(z) = \int_0^1 h(t,\theta)\, dt - \theta \quad \text{where} \quad h(t,\theta) = F(F^{-1}(\theta) + \tau\log(t^{-1} - 1)).$$

We have

$$h(1-t,\theta) = F(F^{-1}(\theta) - \tau\log(t^{-1} - 1)).$$

To simplify notation, let $F^{-1}(\theta) = s$ and $\tau\log(t^{-1} - 1) = a$. Then, $h(t,\theta) = F(s + a)$ and $h(1 - t, \theta) = F(s - a)$. Let us first consider the case $s < 0$ and $a > 0$. We see that

$$F(s+a) - F(s) = \int_s^{s+a} F'(m)\, dm \quad \text{and} \quad F(s) - F(s-a) = \int_{s-a}^s F'(m)\, dm.$$

For any $b > 0$, if $s + b < 0$, then by monotonicity, $F'(s + b) > F'(s - b)$. On the other hand, if $s + b \geq 0$, then $F'(s + b) = F'(-s - b) > F'(s - b)$. In both cases, the right integral is always smaller than the left integral. In other words,

$$F(s+a) + F(s-a) > 2F(s).$$

In fact, the above inequality is also established when $s < 0$ and $a < 0$. Therefore, whenever $s < 0$,

$$\int_0^1 h(t,\theta)\, dt = \int_0^1 \frac{h(t,\theta) + h(1-t,\theta)}{2}\, dt > \int_0^1 F(F^{-1}(\theta))\, dt = \theta.$$

In other words, $\text{Bias}(z) > 0$. The other cases ($s = F^{-1}(\theta) > 0$ and $s = F^{-1}(\theta) = 0$) are similarly proved.

## F    TUNING GUIDANCE FOR TEMPERATURE $\tau$

Theorem 3 suggests that the icdf method converges equally fast as does the Gumbel trick (both on the order of $O(\tau^2)$). On the other hand, the biases depend on $\theta$. Thus, one cannot set temperatures $\tau$, independently of the desired probability $\theta$, to equate the two biases. In practice, $\tau$ is a tunable hyperparameter and a guidance on the tuning range is called for.

To this end, we use a subscript to distinguish the two temperatures—$\tau_\text{g}$ for the Gumbel trick and $\tau_\text{i}$ for the icdf method—and write, based on (12) and (14) and ignoring the high order terms,

$$\frac{\text{Bias}(y_1)}{\text{Bias}(z)} \approx \frac{\tau_\text{g}^2\sigma^2}{\tau_\text{i}^2}r(\theta) \quad \text{where} \quad r(\theta) = \frac{\sqrt{\pi}\theta(1-\theta)(2\theta-1)}{\text{erf}^{-1}(2\theta-1)e^{-(\text{erf}^{-1}(2\theta-1))^2}}.$$

Note that $r(\theta)$ is symmetric around $\theta = \frac{1}{2}$, is concave, attains maximum $\frac{1}{2}$ when $\theta = \frac{1}{2}$, and attains minimum 0 when $\theta = 0, 1$. Hence, if $\tau_g = \tau_i$ and $\sigma = \sqrt{2}$, the bias of the Gumbel trick is (approximately) smaller than that of the icdf method. On the other hand, for a $\sigma > \sqrt{2}$, there exist $\widetilde{\theta}_1 < \widetilde{\theta}_2$ such that $\sigma^{-2} = r(\widetilde{\theta}_1) = r(\widetilde{\theta}_2)$ and that $\mathrm{Bias}(y_1) \gtrapprox \mathrm{Bias}(z)$, whenever $\theta \in [\widetilde{\theta}_1, \widetilde{\theta}_2]$. For example, when $\sigma \approx 2.5$, on the interval $\theta \in [0.01, 0.99]$, the bias of the Gumbel trick is (approximately) greater than that of the icdf method.

Based on the foregoing, a practical guide is to use the same tuning range of $\tau$ for the icdf method as for the Gumbel trick. A small change of $\sigma$ (e.g., $\sqrt{2}$ versus 2.5) will entirely flip the landscape of the bias comparison between the two methods. Because the tuning range is much wider than the change of $\sigma$, for simplicity it suffices to fix $\sigma = 1$.

## G DATA SET DESCRIPTION AND PREPROCESSING

**METR-LA** and **PEMS-BAY.** These are traffic data sets (MIT licensed) used by Li et al. (2018). The former was collected from loop detectors in the highway of Los Angles, CA (Jagadish et al., 2014) and the latter was collected by the California Transportation Agencies Performance Measure System. Both data sets recorded several months of data at the resolution of five minutes. The network graphs are available, which were constructed by imposing a radial basis function on the pairwise distance of sensors at a certain cutoff.

The data sets were originally prepared for forecasting tasks and hence no labeling information exists. We adapt the data for classification. Specifically, we split the time series on the hour, forming hourly windows. We label each window as whether or not it corresponds to rush hour. For proof of concept, we specify 07:00–10:00 and 16:00–19:00 as rush hour and the others non-rush hour. We note that in the original data sets, one of the attributes is time. We remove this attribute to avoid triviality and retain only the speed attribute.

The specification of rush hours may not be highly accurate, but it is a sensible practice to cope with the nonexistence of labeling information. Intuitively, the signal of rush hour comes from reduced traffic speed, but not every location of the network experiences traffic jam. Hence, the diverse traffic patterns inside the same time window under a single label causes nontrivial challenges for local models to discern. Therefore, the need of a global consensus model is justified and it fits well the federated inference scenario.

**PMU-B** and **PMU-C.** These are proprietary data sets coordinately provided by multiple data owners of the U.S. power grid. No personally identifiable information is present. The suffixes B and C indicate the interconnects of the grid. The data sets come with thousands of annotated grid events spanning a period of two years; they form the classification labels. Many variables (attributes) of the grid condition are recorded; we select only the voltage magnitude and the current magnitude, because they appear to be the strongest signals for event detection based on domain knowledge, and also because more data are available for these two variables. The grid topology is not available.

For each event, we select a one-second window from the three-minute window that covers the approximate annotated event time, based on the largest z-score. We retain a sampling frequency of 30Hz, even though some data are 60Hz. Furthermore, a large amount of data are missing in the raw data. We impute the series by using `pandas.DataFrame.interpolate(method = 'linear', limit_direction = 'both')` from the Python `pandas` package. This way, a windowed series is complete if it ever has raw data. Even so, many series are entirely empty, which corresponds to the scenario illustrated by Figure 1.

Classes in these two data sets are rather skewed. For PMU-B, we remove a class that consists of only one data point and for PMU-C, we combine classes that contain fewer than 24 data points into a single class.

## H EXPERIMENT DETAILS

The experiments are conducted on one x86 node of a computing cluster with one K40 Nvidia GPU. The compute node has eight Intel cores and 128GB memory.

For each data set, we perform a 70/10/20 random split for training, validation, and testing, respectively.

For local models, we use LSTM with the same hyperparameters: one hidden layer, hidden dimension = 16, and maximum number of epochs = 200. We pretrain the local models and freeze their parameters afterward.

We train each global model for a maximum of 500 epochs and use early stopping according to the validation loss, with a patience of 50 epochs.

For the GNN global model, we use a 2-layer GCN with skip connections. The hidden dimension is set at 8 and we select the learning rate from $\{0.01, 0.001\}$. For missing data, we impute the node features by using zero.

For the learning of the permutation matrix, we parameterize $K_0$ as $M^2$, where the squaring is elementwise. On data sets METR-LA, PEMS-BAY, and PMU-C, $M$ is initialized using xavier uniform, and for PMU-B it is initialized by an identity matrix with small Gaussian noise. The number of Sinkhorn–Knopp steps is $T = 10$ and the entropy-regularization strength is selected from $\{0, 0.01, 0.1\}$.

For the learning of the consensus graph, the temperature $\tau$ in the Gumbel-softmax method is tuned from $\{0.1, 0.01\}$. The parameter $\tau$ in the icdf method is tuned from the same range, based on the analysis after Theorem 3.

## I  FURTHER EXPERIMENT RESULTS

Extending the last experiment in Section 6, Table 5 summarizes the time and memory consumption during the training of global models on the four data sets. The results indicate that the icdf method is more economic than the Gumbel-softmax method.

Table 5: Time and memory consumption in global model training (five epochs). Time is in seconds and memory is in MB.

|  | METR-LA | | PEMS-BAY | | PMU-B | | PMU-C | |
|  | Time | Memory | Time | Memory | Time | Memory | Time | Memory |
|---|---|---|---|---|---|---|---|---|
| Gumbel trick | 87.89 | 832.38 | 270.52 | 1896.11 | 42.40 | 348.39 | 84.89 | 1119.13 |
| icdf method | 79.69 | 568.24 | 157.93 | 1167.19 | 30.16 | 322.59 | 54.07 | 894.63 |

