# OpenReview forum: "Federated Inference through Aligning Local Representations and Learning a Consensus Graph"
_ICLR.cc/2022/Conference — ICLR 2022 Submitted_

### Official Review · Reviewer_oPhp · 2021-10-31

**Correctness:** 2
**Technical Novelty And Significance:** 3
**Empirical Novelty And Significance:** 2
**Recommendation:** 3
**Confidence:** 4

**Main Review:**

Pros:
(1) The paper proposed an interesting framework to perform "federated inference" to address the scenario which vertical federated learning also studies. This framework can save the expensive coordination among data owners and the central server.
(2) The authors leverage a bunch of techniques to address the hardship of federated inference.
Cons:
(1)	The studied topic is interesting, but the technical ideas have the high risk of leaking privacy of local datasets, due to the upload of new representations of the local data.
(2)	Because of the uploaded representations local datasets, attacks may be performed to infer the local datasets and destroy the privacy protection target.
(3)	In addition, the local representations may be cannot be well learnt by isolating them at first, and thus the alignment and global model training can not be guaranteed as federated learning. In fact, from the results of concatenation method, which has close results as the proposed complex one. The two-stage solution (first representation of local data, and then align representations and global model training) can not sufficiently prove its effectiveness.
(4)	In essence, the technical idea is the first local representations, and then fusion of these representations for decision. Although the authors terms this as “federated inference”, I would like the authors to more care about the “federated” term. In fact, an encryption technique can be adopted for each local data to encode the data and then make the prediction, it is not so clear for the advantage or necessity of federated inference. The authors can further clarify this point.
(5)	Other comments:
(a)The authors should specify methods A and K in Table 2.
(b) There are some similarities between this work and the work of Hu et al. [1] and Chen et al. [2], the authors can introduce a more details of above two papers in the Related Work.
(c) Vertical federated learning also studies such a scenario, I consider the authors should compare the methods of Hu et al. [1] and Chen et al. [2] in the experiments, either in terms of performance or cost.
(d) The current experimental verifications are not so solid and well justified the technical ideas. I would like the authors to consider the weakness/comparisons mentioned in (1)-(4) to confidently prove the proposed method.
[1] Yaochen Hu, Di Niu, Jianming Yang, and Shengping Zhou. FDML: A collaborative machine learning framework for distributed features. In KDD, 2019.
[2] Tianyi Chen, Xiao Jin, Yuejiao Sun, and Wotao Yin. VAFL: a method of vertical asynchronous federated learning. In ICML Workshop, 2020.


**Summary Of The Paper:**

The authors proposed a local-global framework to perform “federated inference” in a less addressed scenario where a datum consists of multiple parts, each of which belongs to a separate owner. Different from federated learning, to perform “federated inference”, joint efforts are required only in inference stage. To enhance the proposed framework, the authors propose two alternative alignment methods and a consensus graph learning among the new representations of different clients. In the experiments, both the theoretical analysis and empirical results are presented to support the effectiveness of the proposed framework. The main technical idea is the isolated learning of representation of local data and then aligning them to make the prediction. But this learning paradigm may heavily hurt the privacy of federated learning, since the new representations may leak the local data, and alignment of multiple representations may also lead to inference attack of these data.

**Summary Of The Review:**

The paper considers a less addressed scenario where a datum consists of multiple parts, each of which belongs to a separate owner and proposes a local-model framework to perform “federated inference” in this scenario.  The main technical idea is the isolated learning of representation of local datasets and then aligning them to make the prediction. But this learning paradigm may heavily hurt the privacy of federated learning, since the new representations may leak the local data, and alignment of multiple representations may also lead to inference attack of these data. Furthermore, the validation and justification of the proposed methods are still not so mature for a top-tier conference.

---

> ### Author Response · Authors · 2021-11-17
> **Rebuttal**
>
> Thank you for the comments. Please find in the following response to each comment.
>
> **RE: Privacy.** Several reviewers raise this concern, but we believe that our work poses no bigger an issue than does the usual FL. We elaborate our points in a separate post titled “On the privacy aspect of this work”. Please find clarifications therein.
>
> **RE: Attack.** We are unsure what attack you refer to. If it means the central server trying to guess the data whose embedding an owner sends, please see the privacy discussion. If you mean other attacks, we will be happy to analyze should you elaborate.
>
> **RE: Point (3).** We are a bit unclear about your point. The concatenation method we compare with is also a two-stage solution. Empirically, the performance of our method is generally better. Take the data set METR-LA for example. In Table 2, the concatenation method (row F) achieves F1 0.824+/-0.006 while our method (model J) achieves 0.839+/-0.006. This is a noticeable improvement. Note that the concatenation method poses a drawback on scalability: the number of global model parameters scales linearly with the number of data owners because of concatenation. On the other hand, the GCN we use will keep the model size unchanged regardless how many data owners there are.
>
> **RE: The “federated” term.** Indeed, data owners can encrypt their local data and predict solely based on the local data; however, each owner likely predicts differently because, for example in our application, each has a different time series recorded at a different geographic location. Hence, a federation of the data owners is needed to collect these different predictions, or as we propose, the local embeddings, to process and aggregate them, and to produce a single label. It is worth noting that a naïve majority voting performs very poorly (see Table 2), and the concatenation method suffers scalability and performs less well than our approach, as mentioned above.
>
> **RE: Methods A and K.** Method A, “common model”, predicts for each time series separately by using a common model, rather than for all time series collectively across owners. This method is for the horizontal FL setting: a single model for all data owners. Method K, “end-to-end”, trains all local models and the global model together, as in the vertical FL setting. Our setting is neither horizontal nor vertical FL; these methods only serve the purpose of checking what results the same data set will achieve in other settings.
>
> **RE: Similarity to [1] and [2].** Indeed, [1] and [2] are vertical FL. Our setting differs from vertical FL in that the local models and the global model are not trained together. Not training them together poses an advantage of little coordination: data owners do not need to agree a time to perform training. (Imagine the headache of asking ten companies to participate collective training.)

---

### Official Review · Reviewer_hLWJ · 2021-11-02

**Correctness:** 2
**Technical Novelty And Significance:** 1
**Empirical Novelty And Significance:** 2
**Recommendation:** 3
**Confidence:** 5

**Main Review:**

Strengths

1. The paper is well-written.
2. The proposed method is practical and technique soundness with insight discussion.
3. The proposed setting is explained well.
4. The main technique contribution comes from the server model design that is to tackle the data heterogeneity, neuron/feature alignment and consensus graph learning.


Weaknesses

1. It will be controversial to name the proposed method as federated inference. Most federated learning methods are designed to serve each user or participant in a distributed system, e.g. smartphone users in cross-device FL, or a banking organisation in cross-silo FL. However, this paper’s setting is to train a global model for server use only while each participant works for data collecting and pre-processing (representation learning). Thus, this setting is more suitable to be categorised as distributed machine learning rather than federated learning.
2. The training of each local model is independent without cross-client collaboration or server-guided coordination. Given the limited resources of each client, it will cause possible practical challenges to training a good model with desired performance.
3.  The experiments are based on datasets for power grid and traffic network with time-series inputs that are unusual benchmarks in FL.
4. Despite the proposed new setting, the proposed method has very limited novelty from a technique perspective. Especially, there is no technical novelty from a graph neural network perspective.


**Summary Of The Paper:**

The paper aims to train a GCN-based multivariate time series classifier by leveraging the pre-processed data (representation per instance) from multiple sources (data owners). The overall model training is decomposed into two steps. First, the client extracts representations using its pre-trained model. Second, the server aligns client-wise representation, and then trains a GCN-based classifier to leverage the structural information across clients.

**Summary Of The Review:**

The paper is a combination between dynamic graph neural networks and federated settings. The proposed method is a practical solution for learning a model in a distributed system with topology information. However, the alignment of federated settings is unconvincing, and the contribution from graph neural networks is also very limited.

---

> ### Author Response · Authors · 2021-11-17
> **Rebuttal**
>
> Thank you for the comments. Please find in the following responses to each weakness point you raise.
>
> **RE: Distributed learning vs federated learning.** We agree that the existing convention for federated learning focuses on serving users/clients, even though a central server exists. This is also part of the reasons we do not consider our setting to be identical to vertical FL. On the other hand, if it is widely accepted, according to Wikipedia and much of the existing literature, that “the main difference between federated learning and distributed learning lies in the assumptions made on the properties of the local datasets, as distributed learning originally aims at parallelizing computing power whereas federated learning originally aims at training on heterogeneous datasets”, then data heterogeneity exactly describes our distinction from distributed learning. Although our setting is similar to vertical FL for ease of exposition (including whether or not the prediction is consumed by the server side or by the client side), neither the server nor the clients has the entire model. Our setting differs from the growing literature of vertical FL in that the server side model and the client side model are not trained end to end, mitigating the challenging coordination among cross-silo clients in practice. (Imagine the headache of asking ten companies to participate collective training.)
>
> **RE: It is challenging to train good local models.** Yes or no; it all depends on the context and application. Take the power grid example we use in the paper. Each data owner has one (or a few) grid sensors. Hence, the data recorded by these sensors are sufficient to train models to classify local events. Global events, on the other hand, need a holistic view of all local predictions; that is exactly the job of the global model. In experiments, we see that majority voting performs very poorly, which partly reflects that local models may not all be correct (in fact, many are incorrect). However, our framework significantly improves over majority voting on the use of a global model with alignment and graph assistance, manifesting the effectiveness of the proposed framework. In addition, we also compare our model with two other baselines which can be seen as training a global model: (A) common model and (K) end2end. The results show that a proper combination of local models will help, even though individual ones are not good enough.
>
> **RE: Unusual benchmarks in FL.** We are not dealing with the usual FL (horizontal FL) and hence a majority of the usual benchmarks are not applicable. On the contrary, we use four data sets that all come from real life and that serve the setting well.
>
> **RE: Novelty.** Our work is novel in the following way: We consider a new setting, which can be solved by straightforward methods. However, we spell out the limitation of the straightforward methods and propose to perform alignment and structure learning to improve them. These two components, including the technical development therein, are key contributions that set up a good reference for follow up work on a similar setting. This paper is not a graph neural network paper and we do not aim to improve graph neural networks.

---

> > ### Comment · Reviewer_hLWJ · 2021-11-30
> > **Thanks for the response.**
> >
> > After reading the response and other reviewers comments, I decide to keep my original score.

---

### Official Review · Reviewer_ePJ1 · 2021-11-02

**Correctness:** 2
**Technical Novelty And Significance:** 2
**Empirical Novelty And Significance:** 2
**Recommendation:** 3
**Confidence:** 5

**Main Review:**

The proposed pipeline is interesting and addresses the relevant problem of heterogeneity in FL. This being said, there are to my opinion important issues that should be addressed:

- The definition of federated inference is not very clear. According to the example depicted in Figure 1, the problem here analyzed consists in FL with heterogeneous data, where heterogeneity arises from different data dimensions within- and between- clients, and different sample size owned by each client. Is this actually the case?

- If this is the case, would simple data standardization and resampling techniques to format the data prior to FL already be sufficient to account for the heterogeneity? My feeling is that a benchmark with simple pre-processing and data imputation approaches is needed to fully appreciate the value of the pipeline proposed in this work.

- The sharing of the data representation extracted from the local models raises important privacy concerns, as it may leak important information about the clients. This is a strong limitation of this work, which should be at least acknowledged in the paper.

- Concerning the sharing of the local representation, the proposed approach could be simply being reformulated as applying dimensionality reduction on the clients’ data (e.g. PCA), and train a centralized model on the top of the latent representations. In what the proposed framework would improve this simple baseline?

- The pipeline requires the further training of centralized models to provide the alignment and consensus graph. It is not clear whether the assessments of Table 2 and 3 are made on the data used for training such a models, or on an independent hold-out dataset.

- Given the complexity of the proposed frameowrk, it would be useful to include a benchmark on controlled scenarios, for example using synthetic data.

- Section 6. The sentence “we smooth out heterogeneity and prepare homogeneous data sets” should be clarified. How is the homogenization performed in a FL setting? Is there any information leaked across clients?


**Summary Of The Paper:**

This work is motivated by the problem of federated inference, which arises when clients own different parts of a datum, and learning must be collectively performed on the complete data representation. This application is presented as an extension of vertical federated learning. The proposed approach is based on the training of local models at the clients’ level. From the local models, an opportune data representation is extracted and shared in a centralized setting, which is then used to perform global inference. To cope with the mismatch of data representations across clients, a centralized alignment  procedure is proposed, which provides the input for a graph neural network encoding the relational structure among clients. Experiments are provided for the specific application of modeling real-life time series datasets from data owners of the US power grid, showing that the proposed local-global analysis pipeline outperforms both local and global training.


**Summary Of The Review:**

The proposed framework is novel and addresses a relevant problem of FL.
The formulation of the paper seems ad-hoc with respect to the proposed application, as well as the related experiments. There are also  concerns on the privacy leaked during the sharing of the local representations.

---

> ### Author Response · Authors · 2021-11-17
> **Rebuttal**
>
> Thank you for the comments. Please find in the following responses to each issue.
>
> **RE: Heterogeneity.** A major source of heterogeneity in our setting is that each data owner may have different data in nature. For example, data owner A’s time series may have length 30 while owner B may have length 60, because of different sampling frequencies. Or, owner A’s time series is 2-dimensional while that for owner B is 4-dimensional, because owners may use sensors manufactured by different companies, recording different attributes of the power grid state. Going beyond, nothing prevents our framework to be applied to more complex cases; for example, owner A has time series, while owner B has only tabular data (e.g., statistics of the otherwise recorded time series).
>
> **RE: Handling of data heterogeneity.** Our framework handles very general settings, including but not limited to all examples mentioned above. When heterogeneity comes from sampling frequency only, as you suggested, resampling and other applicable preprocessing strategies can be used. However, cases can be more complex and a simple preprocessing such as standardization is far from enough. In fact, one may view our local models as preprocessing (either as simple as standardization or arbitrarily complex), which serves an additional purpose of extracting higher level features suitable for classification. Then, a global model consumes all these data/features to predict a single label.
>
> **RE: Privacy.** Several reviewers raise this concern, but we believe that our work poses no bigger an issue than does the usual FL. We elaborate our points in a separate post titled “On the privacy aspect of this work”. Please find clarifications therein.
>
> **RE: Dimensionality reduction.** This issue is related to an early issue, wherein we rebut by posing our local model as a generalization of preprocessing, which dimensionality reduction belongs to. The issues are similar: no matter what local models do, a global model is needed to process the information sent by all local models. Our main contributions lie in the improvement of such processing by the global model instead. For local models, (unsupervised) dimensionality reduction is certainly one choice when feasible; when there are labels, supervised techniques such as what we use as examples are also applicable and may be more effective. In fact, we do not believe that a single model class for local processing works for all applications.
>
> **RE: Tables 2 and 3.** We followed the standard experimentation protocol and performed training/validation/testing splits on the data (see the paragraph “Experiment Setting”): we use the training set to train models and the validation set to tune hyperparameters, while reporting accuracy results on the test set.
>
> **RE: Synthetic data.** While we agree that synthetic data in many occasions are the first indicator of whether an algorithm works, we do not believe there is an undebatable way to synthesize data for the setting this paper is concerned about. For example, in the literature, one did not synthesize a graph with node features to validate if a graph neural network worked, because such models are not based on the distributional assumptions of the graph structure or the feature data. In this work, it is unrealistic to pose assumptions on the arrangement of neurons and the graph; and we do not find a well acceptable way to control these scenarios. To validate the soundness of the approach, instead we perform extensive comparisons with baselines, many of which serve the purpose of ablations, to show the usefulness of alignment and structure learning.
>
> **RE: Clarification of “smoothing out heterogeneity”.** We meant to subsample the time series (if they result from different sampling frequencies) so that all local models deal with the same length of the data. This processing coincides with one of the preprocessing techniques you mentioned earlier. No private information is leaked across clients.

---

### Official Review · Reviewer_ETbu · 2021-11-05

**Correctness:** 3
**Technical Novelty And Significance:** 2
**Empirical Novelty And Significance:** 2
**Recommendation:** 3
**Confidence:** 3

**Main Review:**

The paper made a list of interesting contributions, including formulating a new research problem, developing a complete solution with a few technical innovation. The paper is easy to follow and the results are promising.

However, I have the following concerns of the paper:

1. The problem of federated inference is not well motivated: 1)  For federated learning, privacy concern is a major issue, each individual party only shares the gradient to the center server. In this case, the local party will share an embedding; and this may raise the concern on the data privacy. Especially, it seems the local models are known or being the same from each party (the parameters maybe different). I hope there are some discussion on the privacy aspects. 2) In addition, since the local models are pretrained separately by using the local data, the main problem can be considered as a classification problem with  a number of features f(h_1, h_2, ... h_n), where h_i have the same name of dimension. This naturally leads to model F (local mode+concatenation), which indeed shown to perform quite well against other advanced models (especially on AUC). 3) The applications of the problem seems to be limited; besides the time series data in power grid, are there any other real world applications?

2. Lacking proper baselines: a simple solution seems to be adding MLP (two/three layers) \sigma( W_i^\prime \sigma (W_i h_i)), and then  perhaps concatenate them  for the final aggregated output. In fact, since model F is already performing well, it worth looks into more to see how much it can be improved, and what are the issues of this direction. Those one or two extra layers seem to be able to capture the alignment issue or provide similar functionality.

3. Why the GCN is helpful in this situation? The rational for using GCN is not very clear (Paragraph 1 in Section 5 may need to be extended). And is it a novel contribution to use GCN this way? Since this is a rather simple modification of GCN, I wonder if similar attempt has been done for graph classification (classifying the subgraphs).

4. The technical contributions from Theorem 1 and Theorem 2/3 are very different research problems, and quite disconnected. Are those indeed the main problems for this federated inference problem?

5. The performance of the newly proposed models seems to be relatively small comparing the baseline models, such as model F (table 2).








**Summary Of The Paper:**

This paper introduces a new problem, called federated inference, and proposed a framework of solutions including local representation alignment and learning a consensus graph. The technical contribution mainly comes from a theorem (Theorem 1) on the convergence of learning permutation matrices, and a new approach (icdf) for parameter inference of matrix Bernoulli distribution. The results demonstrate the proposed approaches are useful in improving the prediction accuracy.


**Summary Of The Review:**

Overall, I feel the paper may need some additional work to better connect different technical points, and provide discussions on the necessities of the sophisticated methodology.

===================================
After reading the responses and other reviewers’ comments, I will keep my score.

---

> ### Author Response · Authors · 2021-11-17
> **Rebuttal**
>
> Thank you for the comments. Please find in the following responses to each concern.
>
> **RE: Privacy.** Several reviewers raise this concern, but we believe that our work poses no bigger an issue than does the usual FL. We elaborate our points in a separate post titled “On the privacy aspect of this work”. Please find clarifications therein.
>
> **RE: Comparison with model F in Table 2.** Model F is indeed quite competitive, but it is still subpar to our method. (Note our method is rows I/J rather than row K). A drawback of model F is scalability: the number of model parameters scales linearly with the number of data owners because of concatenation. On the other hand, the GCN we use will keep the model size unchanged regardless how many data owners there are.
>
> **RE: Applications.** Our framework is applicable to all vertical FL applications, which are not limited.
>
> **RE: Another baseline.** Your suggested baseline is similar to model E, in additional model F. We have in fact tested several variants between E and F, but these two are sufficiently representative. The MLP matters little whether it is placed before or after concatenation. What is more important is the concatenation itself as opposed to sum/mean/max pooling. Concatenation works much better, but as we analyze above, it faces a scalability drawback.
>
> **RE: Usefulness of GCN.** GCN can be considered a resolution over the scalability drawback of concatenation and the poor performance of sum/mean/max pooling. It selectively aggregates information from data owners to optimize the final global model prediction. Our treatment is novel; we have not seen the use of GCN in FL when there is not a graph.
>
> **RE: Contributions of Theorems 1--3.** We do not introduce these theoretical results only for the sake of decorating the paper. The two crucial contributions that make the proposed framework work, are the alignment of local embeddings and the use of GCN as the global model. The theorems are natural products when studying the details of these two technical components. For each of them, we compare extensively the possible choices, which eventually lead to the gained improvement over straightforward options. For example, we have compared using GCN versus not and shown that the former works better; we also have compared the Gumbel-softmax approach versus our icdf approach to learn the graph for GCN. The empirical results confirm that our icdf approach yields comparable prediction accuracies while running faster and using less memory, supporting the theorems.
>
> **RE: Improvement over model F.** We believe that there is noticeable improvement. Take the data set METR-LA for example. Model F achieves F1 0.824+/-0.006 while model J achieves 0.839+/-0.006. Note the 0.015 difference. Even counting standard deviations, the two confidence intervals do not overlap. Of course, there is not always such a clear cut; and there is no clear winner between models I and J (as expected, explained in the paper). However, the general improvement is noticeable.

---

### Author Response · Authors · 2021-11-17
**On the privacy aspect of this work**

To all reviewers:

Several of you raise a concern on privacy. We respond to this issue separately in this post.

First, we agree that privacy is an important issue but it is orthogonal to our key contribution here, which focuses instead on a novel method to aggregate diverse data representations generated from heterogeneous local models, which were pre-trained in isolation. Incorporating such localized representations into a coherent universal representation is therefore a challenge in its own right. Second, though our work is not centered on privacy, its modular nature would allow it to straightforwardly incorporate any existing privacy treatments. As local models in our setting were pre-trained in isolation and will not be further altered by our method, it is entirely up to the data owner to sanitize the local representation (not local data) before sending it to the server.

For example, to preserve differential privacy (in order to protect the model against linkage attack), a data owner can perturb their local data representations by calibrated noises (e.g., Gaussian or Laplace noises) which were well established to be effective privacy preservation treatment in the literature. Then, by the post-processing property of differential privacy, we know that any follow-up operations on the sanitized representation would not compromise the sanitization, which guarantees that the outcome of our aggregated model is indeed differentially private. This is indeed similar to how some existing FL works add noise to their gradients before communicating them to the server, but is also more secure in comparison since, unlike the FL setting, representation communication only happens once in our setting.

In addition to differential privacy, the central server may also try to reverse-engineer the data an owner has via their shared representation. This is however a common issue with most existing federated learning algorithms, as long as there is a central server. In the usual (horizontal) FL setting, the central server can reverse-engineer the data from observing differential changes in the model parameters/gradients between iterations, or it can try to do so from observing the data embedded representation as in our case.


For the latter case, however, it is not clear how this can be achieved given that the representation is only shared once in our setting, which is apparently more secure than the iterative gradient sharing in the former case of FL.

At the very least, to the best of our knowledge, there has been no work showing the latter (our setting) is more vulnerable to privacy intrusion than the former (FL setting). Given this, we believe our work bears no bigger a privacy concern than does the usual FL. We will be happy to analyze further this issue in the future but it would be an orthogonal contribution that deserves a separate investigation.

---

### Decision · Program_Chairs · 2022-01-20

**Decision:**

Reject

**Comment:**

The paper proposes to compute local representations on device, which are then shared between clients using an alignment mechanism. Reviewers did appreciate the value of the topic and several contributions, but unfortunately consensus is that it remains below the bar, even after the discussion phase. Concerns remained on privacy and motivational positioning with FL, and lack of simpler baselines, even after the author feedback.

We hope the detailed feedback helps to strengthen the paper for a future occasion.